

# Mid-Cretaceous paleoenvironmental changes in the western Tethys

Cinzia Bottini[1] & Elisabetta Erba[1]

[1]Dipartimento di Scienze della Terra "A. Desio", Università degli Studi di Milano, 20133 Milan, Italy

*Correspondence to:* Cinzia Bottini (cinzia.bottini@unimi.it)

**Abstract**

We present a continuous record of surface water temperature and fertility variations through the latest Barremian – Cenomanian interval (ca. 27 My) based on calcareous nannofossil abundances from the western Tethys. The nannofossil temperature index, calibrated with $TEX_{86}$-Sea Surface Temperatures, suggests that warmest (34-36°C) conditions were reached during Oceanic Anoxic Event (OAE) 1a onset, the Aptian/Albian boundary interval hyperthermals (113, Kilian Level and Urbino Level-OAE 1b), and during a ca. 4My-long phase in the middle Albian. Coolest temperatures (29°C) correspond, instead, to the late Aptian. Generally warm conditions characterized the Albian followed by a progressive cooling trend started in the latest Albian (at the Marne a Fucoidi/Scaglia Bianca Formation transition). Temperate conditions occurred in the Cenomanian with frequent short-term variations highlighted by abundance peaks of cold-water nannofossil species *E. floralis* and *R. parvidentatum*. Mid-Cretaceous surface water fertility was rather fluctuating and mostly independent from climatic conditions as well as from black shales intervals. Intense warming and fertility spikes resulted to be systematically associated only to black shales of OAE 1a and of the Aptian/Albian boundary hyperthermals. The Albian – Cenomanian rhythmic black shales are, in fact, associated with varying long-term climatic/fertility conditions. The similarity of western Tethys climatic/fertility fluctuations during OAE 1a, OAE 1b, the middle Albian and OAE 1d with nannofossil-based records from other basins indicated that these paleoenvironmental conditions were affecting the oceans at supra-regional to global scale.

## 1   Introduction

The mid-Cretaceous was generally characterized by warm climate, with prevailing super-greenhouse conditions, weak latitudinal temperature gradients and absence of ice sheets (e.g. Clarke and Jenkyns 1999; Föllmi 2012; Bodin et al., 2015; Friedrich et al., 2012; O'Brien et al., 2017 and reference therein). Compilation of available Sea Surface Temperatures (SST)s based on $\delta^{18}O$ and $TEX_{86}$ re-calibrated by O'Brien et al. (2017) suggested SSTs between 25 and 40°C through the Aptian – Turonian interval at low latitudinal sites with maximum temperatures reached during the latest Cenomanian – Turonian "thermal maximum". However, the studies performed over the last decades demonstrated that the mid-Cretaceous was not continuously characterized by greenhouse conditions since cooling interludes were identified (e.g. Kemper, 1987; Hochuli et al., 1999; Price, 1999; Herrle and Mutterlose, 2003; Mutterlose et al., 2009; McAnena et al., 2013; Millan et al., 2014; Bottini et al., 2015). Moreover, the ocean-atmosphere system experienced phases of transient, but sometimes prolonged, paleoenvironmental perturbation in relation to widespread ocean anoxia, known as Oceanic Anoxic Events (OAE)s which



included the early Aptian OAE 1a, the early Albian OAE 1b, the latest Albian OAE 1d and the latest Cenomanian OAE 2 (see Weissert and Erba 2004 and Jenkyns, 2010 for review). The majority of paleoenvironmental studies focused on OAEs and little is known about the intervals intervening between these extreme events. This implies that, despite SSTs compilations (e.g. Clark and Jenkyns, 1999; Friedrich et al., 2012; O'Brien et al., 2017), a continuous long-term

reconstruction of paleoecological/paleoclimatic conditions over the Aptian – early Turonian interval is not available.

Important information regarding paleoenvironmental conditions can be gathered from nannofossils since calcareous nannoplankton is extremely sensitive to changes in surface waters chemical and physical conditions. Specifically, decades of studies performed on calcareous nannofossils improved the knowledge about the paleoecology affinity of some species (e.g. Roth and Krumbach, 1986; Bralower, 1988; Premoli Silva et al., 1989a,b; Watkins, 1989; Coccioni et al., 1992; Erba et al.,

1992; Erba, 1994; Herrle and Mutterlose, 2003; Herrle et al., 2003a; Tiraboschi et al., 2009) which were also used to calculate temperature and nutrient indices (e.g. Herrle et al., 2003a,b; Bornemann et al., 2005; Watkins et al., 2005; Browning and Watkins 2008; Tiraboschi et al., 2009; Pauly et al., 2012; Mutterlose and Bottini, 2013; Bottini et al., 2015; Kanungo et al., 2018) over specific intervals of the Cretaceous.

The sedimentary successions of the Umbria-Marche Basin are ideal for long-term studies of paleoenvironmental conditions

since they represent continuous pelagic records through the mid-Cretaceous in the Tethys Ocean. The Aptian-Albian interval corresponds to the Marne a Fucoidi Formation characterized by varicolored marlstones/marly limestones with black shales deposited in the early Aptian (Selli Level – OAE 1a), late Aptian "113 Level", "Kilian Level" at the Aptian/Albian boundary and early Albian (Urbino Level - OAE 1b) (e.g. Coccioni et al., 1987; 1989; Erba 1988; 1992a; Erba et al., 1989). Over the middle-late Albian, the Umbria-Marche Basin was characterized by deposition of rhythmic black shale (Herbert and Fischer,

1986; Premoli Silva et al., 1989b; Erba, 1992a; Galeotti et al., 2003; Tiraboschi et al., 2009). The sedimentation changed in the late Albian as represented by the Scaglia Bianca Formation consisting of whitish pelagic limestones with chert nodules/bands and radiolarian layers. The Scaglia Bianca Formation includes: i) the Pialli Level (Coccioni 2001; Gambacorta et al., 2015), which is the sedimentary expression of OAE 1d, ii) rhythmic black shale and chert layers correlating with the Mid Cenomanian Event I (MCE I) (Coccioni and Galeotti, 2003; Gambacorta et al., 2015) and iii) a ~ 1-

metre thick interval of black shales and radiolarian-rich sands, representing the Bonarelli Level (Bonarelli, 1891) which is the regional sedimentary expression of the latest Cenomanian OAE 2.

Here, we present new quantitative nannofossil data for the western Tethys Ocean (Umbria-Marche Basin, Italy) used to derive surface water temperature and fertility during the late Albian – early Turonian. The new dataset is integrated with the nannofossil data previously collected from the same area for the latest Barremian – late Albian time interval (Tiraboschi et

al., 2009; Bottini et al., 2015) to gain a long-term compilation of temperature and fertility variations at low latitudes through the mid-Cretaceous.

The questions we intend to answer are the following: 1) How the climate and surface water fertility evolved in the western Tethys during the mid-Cretaceous? 2) Were temperature and fertility variations connected and to what degree? 3) How the nannofossil temperature index correlates with other independent proxies of paleotemperature? 4) Are there any limitations in

using calcareous nannofossils as paleoenvironmental tracers? 5) Is there a systematic correlation between black shales and high/low fertility and/or temperature?



## 2    Material and methods

### 2.1    Studied sites

In this work we investigated three nearby sections of the Umbria-Marche Basin (Fig. 1): 1) the Monte Petrano section, spanning the pre-OAE 1d to the early Turonian interval, 2) Le Brecce, covering the OAE 1d and 3) the Furlo sections through the late Cenomanian-early Turonian interval. The sections were selected to cover coeval intervals in order to discriminate any effect of selective diagenesis from primary paleoecological signals.

The studied sequence comprises the Scaglia Bianca Formation that lies above the Marne a Fucoidi Formation (lower Aptian–
upper Albian) and is followed by the Scaglia Rossa Formation (lower Turonian–middle Eocene). Four members were distinguished within the Scaglia Bianca and described by Coccioni and Galeotti 2003).

The Furlo section, 30 m-thick, is located in an abandoned quarry (Beaudoin et al. 1996, Mitchell et al. 2008, Lanci et al. 2010; Gambacorta et al., 2015) in the Furlo Gorge area, 25 km south-east of Urbino (Fig. 1). The section includes the MCE I and OAE 2. The Monte Petrano section covers 80.5 m and it is located along a dirt road not far from the village of Moria
(Schwarzacher 1994, Giorgioni et al., 2012; Gambacorta et al., 2014, Fig. 1) and includes the OAE 1d, MCE I and OAE 2. The Le Brecce section, 20 m-thick, is located 3 km west of the village of Piobbico, at the km 34 point on the state road 257-Apecchiese, close to the Piobbico drill site (Tiraboschi et al., 2009; Gambacorta et al., 2015; Fig. 1). The section covers the interval encompassing OAE 1d.

The stratigraphic framework for the investigated sections is based on integrated lithostratigraphy and carbon-isotope
stratigraphy calibrated with calcareous nannofossil biostratigraphy presented in Gambacorta et al. (2015). Three positive excursions in the $\delta^{13}C$ record identify the OAE 1d, MCE I and OAE 2, respectively. Black shale layers of the Pialli Level (at Le Brecce and Monte Petrano) are restricted to the lower part of the OAE 1d carbon-isotope excursion. The MCE I (at Monte Petrano and Furlo) coincides with a lithological shift to organic-rich black shales and black chert bands alternating with whitish limestones. The $\delta^{13}C$ positive anomaly of OAE 2 (at Monte Petrano and Furlo) is associated with the organic-
rich and carbonate-lean Bonarelli Level, but only the first part of the OAE 2 carbon isotope excursion is preserved since a hiatus at the top of the Bonarelli Level elides most of the "plateau", the "c" peak and the first part of the decrease.  The interval elided corresponds to ca. 340–490 kyr and ~ 320–470 kyr at Monte Petrano and Furlo, respectively (Gambacorta et al., 2015).

### 2.2    Calcareous nannofossil relative abundance

Calcareous nannofossil assemblages were quantitatively investigated under polarizing light microscope at 1250X magnification in smear slides prepared using standard techniques, without centrifuging or cleaning in order to retain the original sedimentary composition. A small quantity of rock was powdered in a mortar with bidistillate water and mounted on a glass slide with Norland Optical Adhesive. A total of 73 smear slides for the Le Brecce section (sampled each 20 cm), 356
smear slides for the Furlo section (sampled each 20 cm) and 171 smear slides for the Monte Petrano (sampled each 50 cm) were investigated. At least 300 nannofossil specimens were counted in each sample and percentages of single taxa were calculated relative to the total nannoflora.



### 2.3 Nannofossil Temperature and Nutrient Indices

Herrle et al. (2003a) proposed two indices based on selected nannofossil species: the Temperature Index (TI) and the Nutrient Index (NI), successfully applied to Aptian and Albian intervals (e.g. Herrle, 2003; Herrle et al., 2003a,b; Tiraboschi et al., 2009; Herrle et al., 2010). Other authors proposed modified version of the TI and NI used in reconstructions through the Aptian (Bottini et al., 2015) and Albian (Tiraboschi et al., 2009).

Here, we adopt the TI and NI of Bottini et al. (2015). The NI includes higher-fertility (*Biscutum constans*, *Zeugrhabdotus*

*erectus*, *Discorhabdus rotatorius*) and lower-fertility (*W. barnesiae*) nannofossil taxa (following Roth and Krumbach, 1986; Premoli Silva et al., 1989a, 1989b; Watkins, 1989; Coccioni et al., 1992; Erba et al., 1992; Williams and Bralower, 1995; Bellanca et al., 1996; Herrle, 2002, 2003; Herrle et al., 2003a; Bornemann et al., 2005; Mutterlose et al., 2005; Tremolada et al., 2006; Tiraboschi et al., 2009). The TI is based on warmer-temperature (*Rhagodiscus asper, Zeugrhabdotus diplogrammus*) and cooler-temperature (*Staurolithites stradneri, Eprolithus floralis, Repagulum parvidentatum*) taxa

(following Roth and Krumbach, 1986; Bralower, 1988; Wise, 1988; Erba, 1992b; Erba et al., 1992; Mutterlose, 1992; Herrle and Mutterlose, 2003; Herrle et al., 2003a; Tiraboschi et al., 2009).

$$TI = (Ss + Ef + Rp) / (Ss + Ef + Rp + Ra + Zd) \ x100 \quad\quad (1)$$

$$NI = (Bc + Dr + Ze) / (Bc + Dr + Ze + Wb) \ x \ 100 \quad\quad (2)$$


Where: $Ss = S.\ stradneri$; $Ef = E.\ floralis$; $Rp = R.\ parvidentatum$; $Ra = R.\ asper$; $Zd = Z.\ diplogrammus$; $Bc = B.\ constans$; $Dr = D.\ rotatorius$; $Ze = Z.\ erectus$; $Wb = W.\ barnesiae$.

The TI and NI of Tiraboschi et al. (2009) were recalculated using Bottini et al. (2015).

### 3 Results

### 3.1 Calcareous nannofossil preservation

Calcareous nannofossils are generally common to abundant in the studied sections, with the exception of the Bonarelli Level that is barren both at Furlo and Monte Petrano. Preservation at Monte Petrano, Le Brecce and Furlo sections is moderate with some evidence of overgrowth. Although in all the three sites, nannofossil assemblages are characterized by relatively

high abundances (> 40%) of *Watznaueria barnesiae* (Figs. 2, 3 and 4) possibly indicative of heavily altered samples (e.g. Thierstein and Roth, 1991), we exclude that nannofossil abundances were controlled by diagenesis since dissolution-prone species (e.g. *D. rotatorius, B. constans, Z. erectus*) are relatively abundant and there is not a systematic correlation between lithology and nannofossil abundance-preservation. We retain that, although lithification and burial process may have altered the micrite composition, the variations in abundance of the species detected preserve a primary signal of oceanic settings.

In the following paragraphs, a description of the major trends of the total nannofossil abundance and of the relative abundance of the taxa used for the calculation of the TI and NI is given for the three investigated sections.

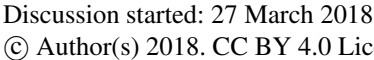



### 3.2 Calcareous nannofossil abundances in the late Albian - early Turonian

In the **Monte Petrano** section (Fig. 2), the total abundance of calcareous nannofossils is comprised between 0.7 and 21.7 (#specimens/field of view) with an average of 9.6 (#specimens/field of view). The Bonarelli Level is barren of calcareous nannofossils. In the interval preceding OAE 1d the total abundance displays relatively lower average (4.8 #specimen/field of view). Higher values (average 12.1 #specimen/field of view) are detected through the OAE 1d up to the base of OAE 2. *W. barnesiae* is the dominant species with mean abundance of 61.6%. Three intervals (0-8 m; 43-50 m; 60-63 m) are characterized by lower *W. barnesiae* abundance (between 30 and 50 %). *Rhagodiscus asper* ranges from 0 to 15% of the total assemblage (mean 3.2%) and the highest values are displayed between -10 and 0 m. *Zeugrhabdotus diplogrammus* ranges from 0 to 6.6% (mean 1%) with the highest peaks between -7 and 8 m. *Staurolithites stradneri* ranges from 0 to 0.4%, and it is encountered from 12.5 to 45 m and 56 to 63 m. *Eprolithus floralis* ranges from 0 to 6.3 % (mean 0.8%) and shows a first increase in relative abundance in the interval between the MCE I and the Bonarelli Level (mean 0.8 %), and a further increase above OAE2 (mean 3.6 %). *Biscutum constans* ranges from 0 to 26.5% (mean 6.4%), *D. rotatorius* from 0 to 9.2% (mean 1.6%), and *Z. erectus* from 0 to 10.5% (mean 1.8%). These three species are more abundant in the interval comprised between the base of the section and the top of OAE 1d. In addition, *Biscutum constans* shows higher values between 60 and 62.5 m, *Z. erectus* displays higher abundances between 38.5 and 46 m, between 56 and 65 m and at the very top of the section. *Discorhabdus rotatorius* and *Z. erectus* show abundances close to zero in the interval above OAE 2.

In the **Le Brecce** section (Fig. 3) the total abundance of calcareous nannofossil is comprised between 0 and 73 (#specimen/field of view) with an average of 15.2 (#specimen/field of view). The highest abundances are registered between 3 and 7 m. *Watznaueria barnesiae* is the dominant species with average abundance of 55.5%. The interval from 0 to 8 m is characterized by relatively higher abundance (average 65%) compared to the interval above (8-20 m) with average abundance of 50%. *Rhagodiscus asper* ranges from 0 to 11% (mean 4.5%), having the highest values within OAE 1d and above it (8-20 m). *Zeugrhabdotus diplogrammus* ranges from 0 to 3.3% (mean 1.35%) and shows higher abundance below OAE 1d. *Eprolithus floralis* and *S. stradneri* are scarce and range from 0 to 1% (mean 0.09%) and from 0 to 0.95% (mean 0.05%), respectively. *S. stradneri* is absent in the interval corresponding to OAE 1d except in two samples (3.4 and 6.6 m). *Biscutum constans* ranges from 0 to 29.7% (mean 15.8%). *D. rotatorius* ranges from 0 to 10.7% (mean 3.1%) and *Z. erectus* from 0 to 5.6% (mean 1.7%). *Biscutum constans* is more abundant from 0 to 3 m and from 11 to 18 m. *D. rotatorius* and *Z. erectus* are more abundant within and above OAE 1d (11 to 20 m).

In the **Furlo** section (Fig. 4) the total abundance of calcareous nannofossils is comprised between 0 and 83 (#specimen/field of view) with an average of 31.69 (#specimen/field of view). The interval corresponding to the Bonarelli Level is barren of calcareous nannofossils. Relatively lower total abundance is identified between 0 and 6 m and between 21.5 and 22.5 m. *W. barnesiae* is the dominant species with a mean abundance of 54%. Only the interval comprised between 22 and 23 m is characterized by much lower average abundance (average 27.25%). *Rhagodiscus asper* ranges from 0 to 7% (mean 0.78%) showing peaks below the MCE I and between 19 and 23 m. *Zeugrhabdotus diplogrammus* fluctuates between 0 and 9.6% (mean 1%) being more abundant in the interval comprised between 0 and 14 m and above OAE 2. *Eprolithus floralis* ranges from 0 to 10.8% (mean 0.3%), highest peaks are displayed above OAE 2. *R. parvidentatum* ranges from 0 to 1.2% (mean 0.05%) with relatively higher values displayed between 11 and 23 m. *Biscutum constans* ranges from 0 to 38.5% (mean 6.7%) showing higher abundance from 0 to 5 m and from 9 to 11 m. *D. rotatorius* ranges from 0 to 2.5% (mean 0.1%) with peaks displayed between 22 and 23 m. *Z. erectus* ranges from 0 to 72.1% (mean: 2.3%) showing the highest values in the lowest part of the section between 2 and 7 m. The latter three species show abundances close to zero in the interval above OAE 2 (27-30 m).





### 3.3 Calcareous nannofossil Temperature and Nutrient Indices (late Albian – early Turonian)

The fluctuations of nannofossil TI and NI are represented in Fig. 5. The Furlo and Le Brecce sections were analysed with a higher sampling resolution (ca. double than Monte Petrano, every 20 cm corresponding to ca. 30 kyrs at Le Brecce and ca. 15-20 kyrs at Furlo, following Lanci et al., 2010) and, consequently, the TI and NI show variations with a higher detail. The purpose of this study is to look at medium- to long-term scale fluctuations, thus very short variations are not commented. To gain comparable resolution (1 m – ca. 100 kyrs) among the datasets, the nannofossil indices of Monte Petrano were
smoothed with a 2 point moving average whilst the TI and NI of Furlo and Le Brecce were smoothed with a 5 point moving average. Relevant peaks (for amplitude and/or stratigraphic correlation) were labelled progressively.

At **Monte Petrano** (Fig. 5) the TI shows relatively cooler conditions before OAE 1d (peaks a and b) followed by a warming starting prior to OAE 1d. General warm conditions are indicated also through OAE 1d characterized by some samples with TI close to zero (corresponding to highest temperature) and a few minor cooling intervals (c, d, e?). The interval comprised
between the top of OAE 1d and MCE I is characterized by average TI shifted towards slightly cooler conditions (peaks f, g, h, i, l). The MCE I is associated to transient warming followed by increasing TI values (peak m), suggestive of the onset of a progressive cooling interval with frequent oscillations in TI values. Three cooler interludes are identified over this interval (peaks n, o, p) alternated with warmer phases. A marked warming is detected just prior to OAE 2. All samples in the Bonarelli Level are barren of calcareous nannofossils and therefore the TI cannot be calculated. In the early Turonian, above
the hiatus eliding the earliest Turonian "thermal maximum" (Gambacorta et al., 2015), the TI is indicative of cooler conditions (peaks q and r).

At Monte Petrano the NI is suggestive of higher surface water fertility characterizing the interval preceding OAE 1d (peaks a to d). A decrease of the NI just precedes the onset of OAE 1d and characterizes the early phase of the event. Relatively higher fertility is detected during the middle and upper part of OAE 1d (peaks e?, f). The interval comprised between OAE
1d and MCE I is marked by a decreasing trend of the NI indicating progressively lower surface-water fertility that remained suppressed for most of the Cenomanian. A moderate increase of the NI started in the NC10b nannofossil subzone (peak h?) and values remained almost constant up to ca. 60 m except for minor peaks (i?, l, m?, n?) including the one detected during MCE I. A shift towards higher NI (peak p) is identified in a discrete interval below OAE 2 (60 - 62 m) followed by an interval of low NI and, again, by a rise just prior to OAE 2 (peak q). All samples in the Bonarelli Level are barren of
calcareous nannofossils and therefore the NI cannot be derived. The interval above the Bonarelli Level shows low NI except for a minor peak at the very top of the studied interval (peak r?).

The interval before OAE 1d at **Le Brecce** (Fig. 5), shows relatively cooler conditions (peak b) although followed by some samples with lower TI prior to OAE 1d. Two minor peaks towards relatively cooler temperature mark the early phase of
OAE 1d (peaks c, d) while the rest of the event is characterized by warm conditions with most samples having TI equal zero. A very minor cooling peak (e) is evidenced close to the end of OAE 1d. Warm conditions mark the interval post OAE 1d.

In the Le Brecce section the NI is suggestive of higher surface water fertility in the interval preceding OAE 1d (peak d). A decrease in fertility marks the onset of OAE 1d. An increase of the NI is detected in the middle and upper part of OAE 1d (peaks e, f). Here, the NI is more fluctuating with alternated samples having relative higher and lower NI values. The interval
following OAE 1d is marked by a progressive decrease of NI values.



In the **Furlo** section (Fig. 5), the lower interval preceding the MCE I is characterized by a warming trend (although with three cooling spikes h, i, l) reaching warmest conditions during MCE I onset although samples with relatively higher TI are encountered. A progressive temperature decrease is registered at the end and after MCE I (peaks m, n). Fluctuating TI values

marks the interval between the MCE I and OAE 2 with four subsequent cooler intervals (peaks o1, o2, p1, p2). Prior to OAE 2 all samples reached TI=0 marking intense warming. Samples in the Bonarelli Level are barren of calcareous nannofossils and therefore the TI cannot be calculated. The TI record above the Bonarelli Level, above the hiatus eliding the earliest Turonian "thermal maximum" (Gambacorta et al., 2015), is suggestive of cooler conditions (peaks q, r) although a few samples shift towards very low/zero values of the TI suggestive of short-lived warming spikes within a much cooler interval.

The NI is suggestive of higher surface water fertility over the interval preceding the MCE I (peaks h, i) whose onset is characterized by higher NI (peak l) followed by a decrease of fertility reaching lower values soon after MCE I. The overlying Cenomanian interval is characterized by relatively higher NI (peaks m, n, o) but displaying a decreasing trend. Below OAE 2 a distinct fertility phase is detected (peak p); this is followed by an interval of low NI until a peak just preceding OAE 2 (peak q). All samples in the Bonarelli Level are barren of calcareous nannofossils and therefore the NI

could not be used. The interval above the Bonarelli Level shows low NI values.

### 3.3.1     Comparison of the TI and NI in the three studied sections

The smoothed TI and NI calculated in the three studied sections are compared (Fig. 5) to i) detect if similarities/differences in the main trends reflect primary signals and ii) understand if records with different resolution highlight the same number

and amplitude of fluctuations.

The TI and NI records obtained across the OAE 1d interval at Le Brecce and Monte Petrano show many similarities in trends and values. Concerning surface water fertility, at both sections, there is a decrease in the NI starting prior to OAE 1d followed by an increasing trend initiating in the middle of the Pialli Level (peak e). The highest NI values are reached in the terminal part of OAE 1d (peak f). Higher NI values are determined mainly by abundant *B. constans* in both sections (20-30

%). The TI indicates relatively cooler conditions prior to OAE 1d at either Monte Petrano and Le Brecce sections with a relative warming just preceding OAE 1d. The relative cooling at the onset (peak c) is more pronounced at Le Brecce whilst other fluctuations are rather similar. General warm conditions during OAE 1d are identified in both sections derived from higher abundances of *R. asper* (warm-water species) combined with very rare (≤ 1 %) cold-water species.

Considering Furlo and Monte Petrano, the TI shows similar trends although at Monte Petrano the TI has slightly higher

values (cooler conditions) in the interval postdating the MCE I due to relatively higher abundance of *E. floralis*. The sequence of warmer and cooler phases detected from the MCE I up to the onset of OAE 2 are related to alternated peaks of *R. asper* and *E. floralis/R. parvidentatum* (cold-water species). *R. parvidentatum* is found, although rare, only at Furlo. The main difference in the two TI records regards the more pronounced fluctuations (o1, o2; p1, p2) at Furlo probably due to the lower resolution adopted at Monte Petrano (peaks o and p). In both studied sections, the TI indicates a warming starting 3 m

below the Bonarelli Level (ca. 400 kyrs, Lanci et al., 2010), whilst the interval above the Bonarelli Level, corresponding to the post-early Turonian "thermal maximum", is characterized by abundant *E. floralis* suggestive of cooler temperatures.

More pronounced differences are evidenced in the NI curves of Monte Petrano and Furlo. In particular, the NI calculated at Furlo is 2 to 3 times higher than at Monte Petrano due to more abundant *B. constans* and *Z. erectus* at Furlo. Such a



difference in the smoothed NI values exists up to the middle part of the nannofossil zone NC11, but from the middle part of NC11 to the base of OAE 2, the two NI curves show similar values. A fertility spike (peak p) preceding OAE 2 of ca. tot 400 kyrs (Lanci et al., 2010) coincides with the cooler interlude traced by the TI (peak $p_2$). The NI spike derives from higher abundances of *B. constans*, *Z. erectus* and *D. rotatorius* while *W. barnesiae* (oligotrophic species) shows a minimum in abundance. Low NI values characterize both sections in the interval above the Bonarelli Level.

The comparison of the three studied sites suggests that the relative abundances of analysed nannofossil species and the TI and NI display similar and coeval fluctuations. This indicates that a primary signal is recorded by nannofossils. We interpret the differences in the NI of Furlo vs Monte Petrano to depend from primary factors. We notice, in fact, that only fertility indicator species differ in relative abundance, whilst the other analysed species have similar. The higher abundances of mesotrophic species at Furlo is, therefore, considered to reflect a relatively different setting at Furlo hypothesized to be

located closer to the coast and, thus, interested by a higher input of nutrients from the continent as also suggested by sedimentological data (Gambacorta et al., 2016).

## 4    Discussion

The nannofossil TI and NI derived for the Albian-Cenomanian time interval extend the record previously collected for the

latest Barremian – Aptian (Bottini et al., 2015) and provide, for the first time, a complete and continuous reconstruction of paleoenvironmental conditions through the mid-Cretaceous of the western Tethys Ocean. We built a composite record (Fig. 6) which includes, from bottom to top: 1) the Cismon Core for the latest Barremian – early Aptian (up to the top of the Selli Level); 2) the Piobbico Core for the early Aptian (from the top of the Selli Level) – late Albian; 3) Le Brecce section for the late Albian; 4) the Monte Petrano section for the late Albian – Cenomanian.


### 4.1    Long-term changes in surface water fertility

The NI (Fig. 6) indicates low fertility in the latest Barremian – early Aptian interval (nannofossil Zone NC6) except for relative increase of fertility at the nannoconid decline and at the onset of OAE 1a. Mesotrophic conditions are reconstructed through the late Aptian (nannofossil Zone NC7) with a temporary return to oligotrophy during the *N. truittii* acme interval.

In the Aptian/Albian boundary interval, surface water fertility was again relatively low despite short-lived peaks in fertility marking the Kilian and Urbino Levels and, partially, the Monte Nerone Level. The middle Albian (nannofossil Zone NC9) was characterized by overall mesotrophic conditions comparable to those detected for the late Aptian.

At the beginning of the late Albian, in correspondence of the lithostratigraphic change from the Marne a Fucoidi to the Scaglia Bianca Formation, a relatively short interval of oligotrophic conditions is followed by an increase in fertility,

interrupted by a temporary drop associated to the Pialli Level (onset of OAE 1d). Maximum NI values were reached in correspondence of the end of OAE 1d. A return to relatively low fertility was derived for the early Cenomanian (upper part of nannofossil Zone NC10), followed by intermediate conditions through the rest of the Cenomanian. However, differences were evidenced at the investigated localities: at Monte Petrano the NI shows medium to low fertility, whereas at Furlo conditions were mesotrophic, possibly because this location was more proximal and under the influence of higher nutrient





input due to run-off (Gambacorta et al., 2016). By late Cenomanian time (upper part of nannofossil Zone NC11*) a discrete
        interval of higher fertility was identified ca. 400 kyrs prior to the onset of OAE 2.

### 4.2    Long-term changes in surface water temperature

        The composite TI obtained from individual sections (Fig. 5) was modified taking into account the relative abundance of the
warm-water species *R. asper* for the intervals marked by highest nannofossil-paleotemperature (TI = 0). Such modification
        allows the evaluation of potential fluctuations under excess warming when the TI reaches the full-scale. The modified TI (m-
        TI) (Fig. 6) was here applied also to the western Tethyan records previously reconstructed by Tiraboschi et al. (2009) and
        Bottini et al. (2015) for the Albian and the Aptian, respectively, to derive a composite Aptian – Cenomanian curve.

        The m-TI shows relatively warm conditions through the latest Barremian – earliest Aptian with a rapid warming event at
OAE 1a onset, followed by a return to pre-OAE 1a paleotemperatures through the rest of the Selli event. It is, however, of
        remark that *E. floralis* globally appeared immediately after the end of OAE 1a. Consequently, warm TI paleotemperatures
        obtained for the latest Barremian – earliest Aptian interval are, possibly, overestimated. Soon after OAE 1a, the m-TI shows
        a cooling persisting through the late Aptian, with the lowest temperatures reached immediately after the *N. truittii* acme
        interval. A progressive increase in paleotemperatures characterized the latest Aptian – earliest Albian (nannofossil Zone
NC8) and was marked by hyperthermals detected in association to specific black shale levels (Kilian equivalent and Urbino
        Level). The highest paleotemperature - similar to that of OAE 1a onset – was reached within OAE 1b. The m-TI record of
        the middle Albian (nannofossil Zone NC9) underlines a prolonged warming phase with conditions similar to those
        characterizing the onset of OAE 1a and OAE 1b, and three cycles are recorded (Fig. 6). In the late Albian – Cenomanian
        interval, the m-TI shows intermediate conditions with ample fluctuations becoming more frequent in the middle to late
Cenomanian between the MCE I and OAE 2.

        The relationships between the nannofossil TI and other paleotemperature proxies were previously discussed for the Aptian-
        Albian interval (Tiraboschi et al., 2009; Bottini et al., 2015). A limited agreement is observed between the $\delta^{18}O_{(bulk)}$ record
        and the m-TI, although temperature trends appear quite similar across OAE 1a, in late Aptian-early Albian hyperthermals,
        and during OAE 1d (Fig. 6).

TEX$_{86}$-derived SSTs from the Proto-North Atlantic (McAnena et al., 2013) were used to calibrate the average nannofossil TI
        obtained from DSDP Site 463 and Piobbico Core through the late Aptian (Bottini et al., 2015). Here, a similar approach is
        applied using the m-TI and available TEX$_{86}$-SSTs at low latitudes (Fig. 6), namely: 1) DSDP Site 463 Mid-Pacific
        Mountains for OAE 1a (Dumitrescu et al., 2006), 2) DSDP Site 545 Proto-North Atlantic for the late Aptian (McAnena et
        al., 2013), 3) and ODP Sites 1258-1259 Demerara Rise for middle Albian-Cenomanian (Forster et al., 2007). Following
O'Brien et al. (2017) all SSTs were re-calculated with the calibration proposed by Kim et al. (2010) for oceans with SST >
        15°C. Two tie points were used to calibrate the m-TI curve with TEX$_{86}$-SSTs during OAE 1a and the early late Aptian, prior
        to the *N. truittii* acme. In fact, these intervals were characterized by similar temperatures worldwide (e.g. Schouten et al.,
        2003; Dumitrescu et al., 2006; Mutterlose et al., 2010, 2014) and equal average TI values for the Tethys and Pacific Oceans
        (Bottini et al., 2015).

For the late Aptian, the nannofossil TI indicates Tethyan temperatures warmer than in the Pacific Ocean (Bottini et al., 2015)
        and we assumed that TEX$_{86}$-SSTs in the Proto-North Atlantic were lower than in the western Tethys (Fig. 6) as indicated by
        higher relative abundance of the cool-water boreal taxon *R. parvidentatum* in the Proto-North Atlantic (McAnena et al.,



2013) than at Piobbico (Fig. 6). The calibrated m-TI shows an average paleotemperature of ca. 33°C in the interval preceding OAE 1a and ca. 36°C at the onset of OAE 1a. It might however be, that m-TI paleotemperatures of the Barremian/Aptian

boundary interval are overestimated. The absence of the cold-water species *E. floralis* could have in fact altered the nannofossil results as discussed above. Considering that the nannofossil TI reconstructed from Tethyan and Pacific records (Bottini et al., 2015) for the latest Barremian – earliest Aptian time interval provided comparable results and that a ca. 8°C increase was derived at the onset of OAE 1a at DSDP Site 463 (Ando et al., 2008), we suppose that m-TI SSTs across the Barremian/Aptian boundary are overestimated by 3-4°C. The coolest conditions of the middle-late Aptian corresponded to

paleotemperatures of ca. 29°C followed by an increase up to 32°C across the Aptian/Albian boundary with hyperthermal spikes of 31°C in the 113 Level, 34°C in the Kilian Level and 36°C in the Urbino Level. Across the middle-late Albian, the m-TI from western Tethys and TEX$_{86}$-SSTs from Demerara Rise (ODP Sites 1258) displays a long-lasting warming phase with temperatures fluctuating between 34°C and 36°C. A 2°C decrease of paleotemperatures was recorded for the late Albian – early Cenomanian (33°C on average) with oscillations between 32°C and 34°C. Nannofossil m-TI derived

paleotemperatures suggest a further cooling starting before the MCE I with values fluctuating around 32°C.

In the late Cenomanian, TEX$_{86}$-SSTs were ca. 2 to 4 °C higher with respect to m-TI derived values. Indeed, *R. parvidentatum* was found only in the western Tethys, thus providing evidence for cooler conditions relative to the equatorial Demerara Rise (Hardas et al., 2012) where TEX$_{86}$-SSTs were calculated (Forster et al., 2007). The m-TI based paleotemperature shows a warming trend prior to OAE 2, but the absence of calcareous nannofossils in the Bonarelli Level

hampers climatic reconstructions across the Cenomanian/Turonian boundary.

The comparison of paleotemperatures reconstructed in this work with average low latitude TEX$_{86}$-SSTs compiled by O'Brien et al. (2017), shows similar trends and temperature values for the Aptian and earliest Albian (Fig. 7). Middle Albian TEX$_{86}$-SST record is instead limited to restricted intervals (middle part of nannofossil subzone NC9a and NC9b) but shows comparable range of temperatures to the m-TI-SSTs. The m-TI also indicates similar SSTs to compiled TEX$_{86}$-SSTs for the

early and middle Cenomanian. The two paleotemperature proxies differ instead across the late Cenomanian, when TEX$_{86}$-SSTs were 2 to 4 °C higher compared to calibrated m-TI.

**4.3     mid-Cretaceous environmental changes in the western Tethys: paleoceanographic implications**

The temperature and fertility variations reconstructed for the latest Barremian to Cenomanian time interval (ca. 27 Ma), are discussed to derive causal or casual relationships with black shale deposition. It is, instead, not the purpose of this work to investigate the causes and processes which have induced the climatic and oceanographic changes identified.

The Temperature and Nutrient indices show long-term variations that are not systematically changing in phase or antiphase, suggesting that the two parameters were mostly independent one from the other. The record indicates, in fact, that intervals

of higher fertility occurred during either warmer or cooler climatic conditions. An exception is represented by the hyperthermals (onset of OAE 1a, Kilian and Urbino Levels) which consistently corresponded to higher fertility. Evidence for combined higher fertility and higher temperatures at the beginning of OAE 1a and across OAE 1b was additionally provided by nannofossil TI and NI at DSDP Site 463 (Bottini et al., 2015), and from the Vocontian Basin (Herrle et al., 2003a,b; Herrle 2003) and the Black Nose (western Atlantic Ocean, Browning and Watkins 2008), respectively. The prolonged warm

phase (ca. 4 My) identified in the middle Albian was also associated with long-lasting higher fertility. Calcareous



nannofossil and oxygen isotope data (Kanungo et al., 2018) from the Gault Clay Formation (UK) show, similarly to the Tethys, an increase in temperature and fertility (Figure 7) in the middle part of nannofossil subzone NC9a (middle Albian). Paleoenvironmental conditions across OAE 1d point to warm climate and low fertility followed by lower temperature and higher fertility (Figure 7). Similar paleoetemparture and fertility fluctuations are recorded at supra-regional scale by

nannofossil TI and NI from the Vocontian Basin (Bornemann et al. 2005) and NI from the Black Nose (western Atlantic Ocean, ODP Hole 1052; Watkins et al 2005). We therefore conclude that some of the climatic/fertility changes documented in the western Tethys were affecting the oceans at least at supra-regional scale.

In addition to OAEs, the Albian – Cenomanian record from the western Tethys is characterized by rhythmic black shales (Tiraboschi et al., 2009; Gambacorta et al., 2015) associated with varying long- and short-term climatic/fertility conditions.

Specifically, in the early – middle Albian, anoxic sediments occurred under temperate and lower fertility conditions. But in the middle – late Albian, rhythmic black shales were associated with warm and generally more fertile surface waters. Lower – middle Albian black shales did not correspond to significant changes in surface water fertility or temperature, whereas intense warming and less fertile surface waters were reconstructed for middle – upper Albian black shales (Tiraboschi et al., 2009). These data suggest that fertility was not the primary factor controlling the deposition of rhythmic black shales which

were instead related to water stratification induced by lowered salinity (early – middle Albian) and warming (middle – late Albian).

Middle to late Cenomanian black shales (between the MCE I and the Bonarelli Level) were mostly distributed during an interval of average lower fertility and temperate conditions, partly similar to those identified in the early Albian. In our record, the MCE I corresponds to the onset of more frequent climatic fluctuations evidenced by pulses of abundant cold-

water taxa *R. parvidentatum* and *E. floralis*. Previous works identified in the MCE I a turning point in climatic/oceanographic regime leading to different paleoenvironmental conditions during the mid-Cenomanian (e.g. Erbacher and Thurow, 1997; Coccioni and Galeotti, 2003; Voigt et al., 2004; Friedrich et al., 2009; Zheng et al., 2016). Across middle-late Cenomanian black shales, short-term changes in nannofossil assemblages do not evidence a systematic increase in temperature or fertility during black shale deposition. These rhythmic anoxic intervals were suggested to derive

from pulsed water stratification interrupted by intervals of bottom current activity which temporarily restored oxygenated conditions (Gambacorta et al., 2016). A peculiar 200 kyr-long interval, preceding by ca. 400 kyr the Bonarelli Level, was marked by closely-spaced black shales in a discrete interval of higher fertility and cooler conditions.

As discussed by Giorgioni et al. (2015), the late Albian change in sedimentation towards micritic pelagic limestones recognized in the Tethys and North Atlantic Oceans infers a major oceanic circulation rearrangement possibly derived from

much better connection between different oceanic basins and developed circulation mode. In the deep western Tethys, this time coincided with the shift from the Marne a Fucoidi varicolored marlstones-black shales to the Scaglia Bianca whitish limestones. The reconstructed TI-NI records (this work) indicate that such lithostratigraphic change corresponded to the end of the middle Albian warm phase and progressively decreasing temperatures. Calcareous nannofossil assemblages, therefore, provide evidence for a new paleoceanographic regime under improved interbasinal connections and long-term sea level rise

(e.g., Poulsen et al., 2003; Giorgioni et al., 2015). We notice that the upper Albian – lower Cenomanian Scaglia Bianca Formation is characterized by two intervals of red-colored limestones. Mid-Cretaceous red beds were interpreted to reflect good oxygenation possibly induced by the inflow of colder and, thus, more oxygenated bottom waters (Hu et al., 2005). The presence of red beds also in correspondence of the late Aptian cooling and during temporary -and relative- drop in temperatures during the middle Albian, may further suggest that cooler climate influenced ocean oxygenation by promoting

better oceanic circulation.





## Conclusions

1. Quantitative analyses of nannofossil assemblages resulted in the first complete record of surface water temperature and fertility variations through the latest Barremian –Cenomanian interval (ca. 27 My) in the western Tethys (central Italy).

2. A modified nannofossil TI (m-TI), which takes into account the relative abundance of warm-water species *R. asper,* is proposed to estimate the extent of the warming when the TI reaches the full scale (= zero), maximum warmth. The calibration of the m-TI with $TEX_{86}$-SSTs allowed the assignment of SSTs values to the m-TI. Warmest conditions in the Aptian (ca. 36°C) were identified at the onset of OAE 1a. The middle – late Aptian was, instead, marked by the coolest conditions (ca. 29°C) of the entire studied interval. Hyperthermals were recognized across the Aptian/Albian boundary interval in the 113 Level (31°C), Kilian Level (34°C) and Urbino Level-OAE 1b (36°C) which was the last and most intense of the hyperthermals with temperatures similar to those estimated for the onset of OAE 1a. Similar high temperatures were reached in the middle Albian (34°C-36°C) that corresponds to the most prolonged (ca. 4 Myr) warm phase of the studied interval. Temperate climate was instead reconstructed for the early Albian and the Cenomanian although, in the Cenomanian, temperatures were strongly fluctuating being comprised between 32°C and 34°C.

3. SSTs in the western Tethys resulted to be similar to SSTs from other oceanic basins and different latitudes during the early Aptian, in the middle Albian and in the early-middle Cenomanian. During the late Aptian, the western Tethys SSTs were instead ca. 1°C warmer than in the Proto-North Atlantic whilst in the late Cenomanian they were 2-4°C cooler with respect to paleotemperatures of the equatorial Atlantic Ocean.

4. The Albian – Cenomanian rhythmic black shales were associated with varying long- and short-term climatic/fertility conditions: i) temperate and lower fertility in the early – middle Albian, ii) warm and generally more fertile surface waters in the middle – late Albian, iii) lower fertility and temperate conditions in the middle to late Cenomanian.

5. A peculiar black shales interval preceding by ca. 400 kyr the Bonarelli Level occurred during a phase of higher fertility and cooler conditions.

6. The shift from the Marne a Fucoidi to the Scaglia Bianca Formation coincided with the end of the middle Albian warm phase and onset of pelagic limestone deposition, reflecting a new paleoceanographic regime with possibly improved inter-oceanic connections under rising sea level.

7. The MCE I corresponded, instead, to the onset of more frequent climatic fluctuations and rhythmic black shales suggestive of alternated phases of water stratification and enhanced bottom current activity.

8. All intervals represented by red-colored lithologies, such as in the late Aptian, middle Albian and early Cenomanian, correspond to cooler conditions, possibly influencing the degree of oxygenation at the sea-floor.

*Acknowledgements*. C. Bottini was funded through SIR-2014 (Ministero dell'Istruzione, dell'Università e della Ricerca– Scientific Independence of young Researchers) to C. Bottini.

## Appendix A: Taxonomy

Calcareous nannofossils cited in this work:

*Biscutum* Black in Black and Barnes, 1959



*Biscutum constans* (Górka 1957) Black in Black and Barnes, 1959

*Discorhabdus* Noël, 1965

*Discorhabdus rotatorius* (Bukry 1969) Thierstein 1973

*Eprolithus* Stover, 1966

*Eprolithus floralis* (Stradner, 1962) Stover, 1966

*Repagulum* Forchheimer, 1972

*Repagulum parvidentatum* (Deflandre and Fert, 1954) Forchheimer, 1972

*Rhagodiscus* Reinhardt, 1967

*Rhagodiscus asper* (Stradner, 1963) Reinhardt, 1967

*Staurolithites* Caratini, 1963

*Staurolithites stradneri* (Rood et al., 1971) Bown, 1998

*Watznaueria* Reinhardt, 1964

*Watznaueria barnesiae* (Black, 1959) Perch-Nielsen, 1968

*Zeugrhabdotus* Reinhardt, 1965

*Zeugrhabdotus diplogrammus* (Deflandre in Deflandre and Fert, 1954) Burnett in Gale et al., 1996

*Zeugrhabdotus erectus* (Deflandre in Deflandre and Fert, 1954) Reinhardt, 1965

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

**Figures**





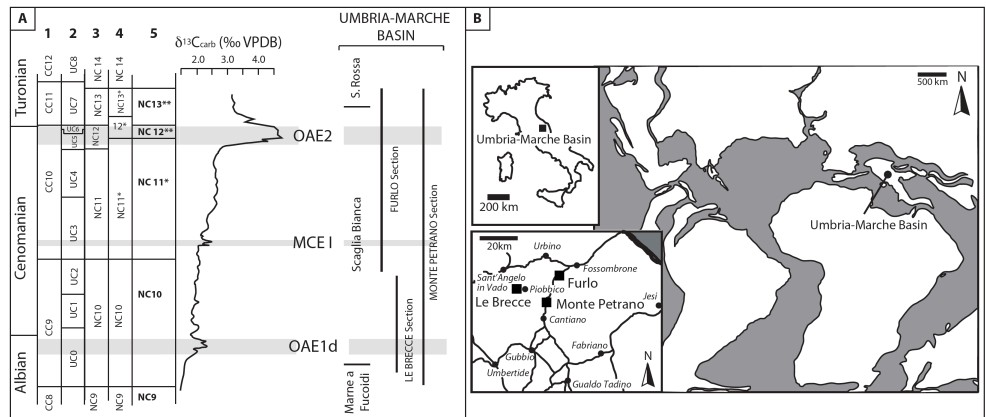

**Figure 1: A) Stratigraphic ranges of the studied sections plotted against nannofossil biostratigraphy and chemostratigraphy of the late Albian – early Turonian time interval modified after Gambacorta et al. (2015). Nannofossil zones after 1: Sissingh (1977), 2: Burnett (1998), 3: Roth (1978), 4: Bralower et al. (1995), 5: Tsikos et al. (2004), Gambacorta et al. (2015). FO: first occurrence, LO: last occurrence. B) Present-day and paleogeographic location of the studied sections, modified after Gambacorta et al. (2015).**

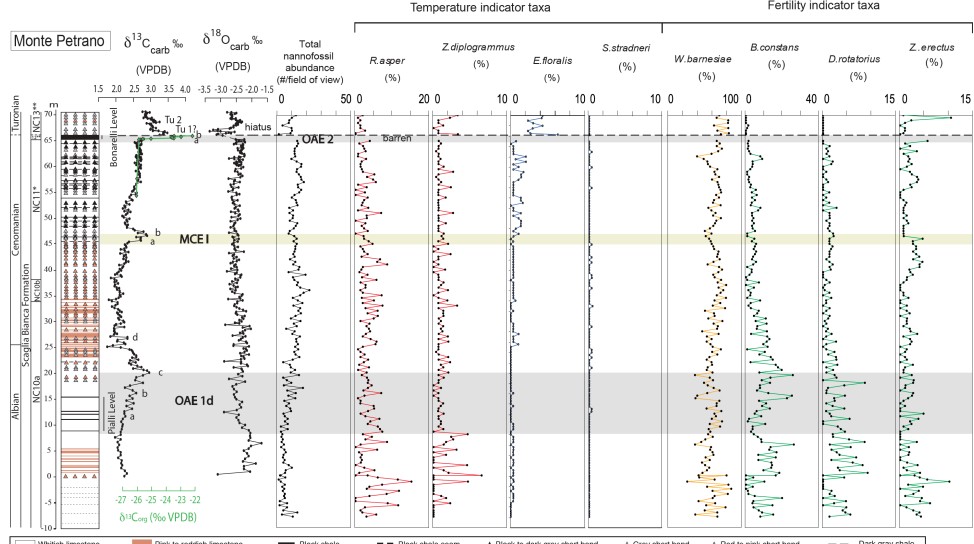

**Figure 2: Monte Petrano: total calcareous nannofossil abundance, relative (%) abundance of temperature and fertility indicator nannofossil species. The OAE 1d and OAE 2 are indicated by a grey band, the MCE I is represented by a yellow band. The stratigraphic framework (litho-, bio-, chemo-stratigraphy) is from Gambacorta et al. (2015).**



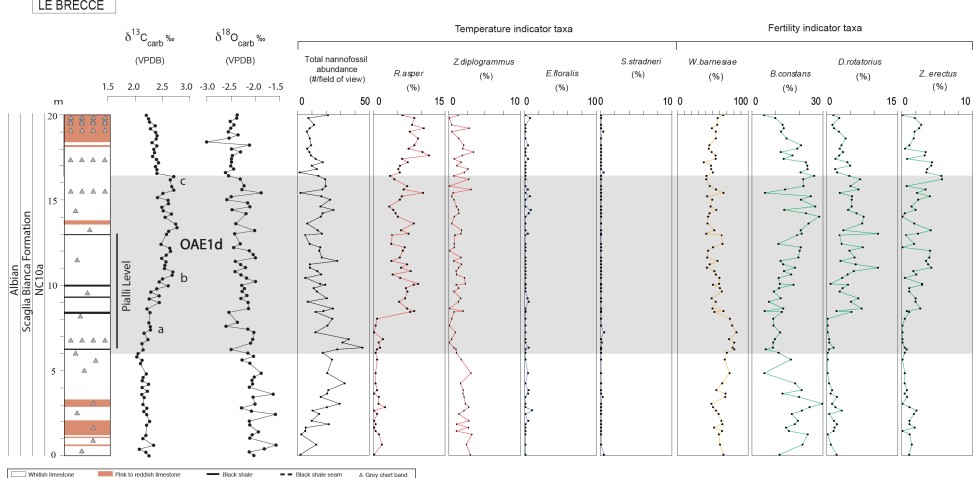

**Figure 3:** Le Brecce: total calcareous nannofossil abundance, relative (%) abundance of temperature and fertility indicator nannofossil species. The OAE 1d is indicated by a grey band. The stratigraphic framework (litho-, bio-, chemo-stratigraphy) is from Gambacorta et al. (2015).

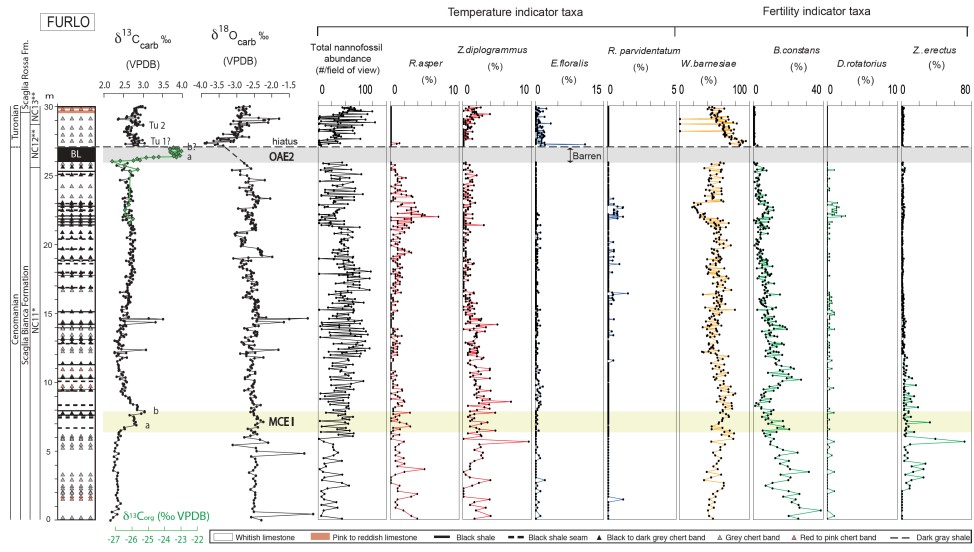

**Figure 4:** Furlo: total calcareous nannofossil abundance, relative (%) abundance of temperature and fertility indicator nannofossil species. The OAE 2 is indicated by a grey band, the MCE I is represented by a yellow band. The stratigraphic framework (litho-, bio-, chemo-stratigraphy) is from Gambacorta et al. (2015).





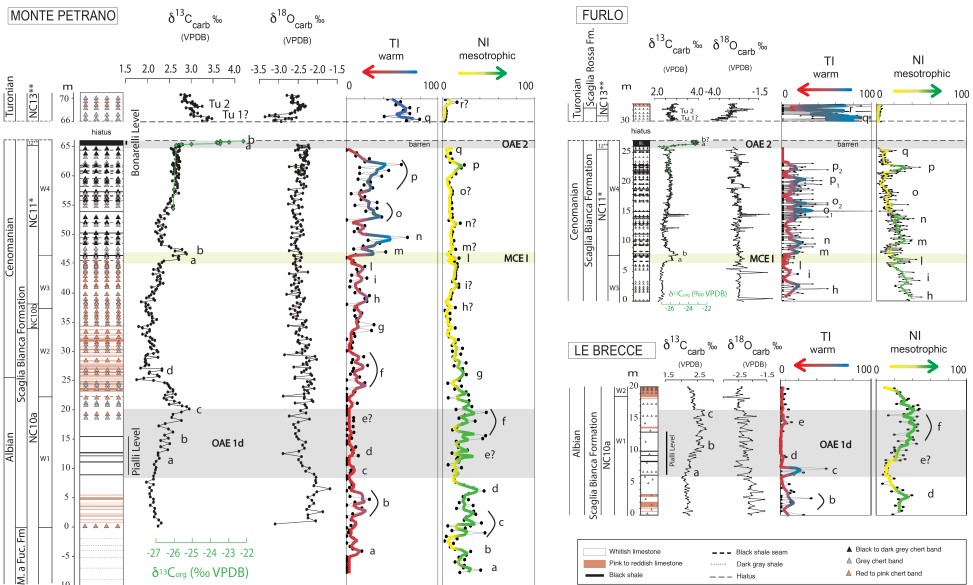

**Figure 5: Nannofossil Temperature (TI) and Nutrient (NI) Indices for Monte Petrano, Furlo and Le Brecce sections. TI and NI are**

**reported point by point (in black), and smoothed (TI=blue-red; NI = green-yellow) with a 3pt moving average (Monte Petrano)**
**and 5pt moving average (Furlo and Le Brecce). Lower values of the TI indicate higher temperatures and vice versa. Higher values**
**of the NI indicate higher surface water productivity and vice versa. Litho-, bio-, chemo-stratigraphy is from Gambacorta et al.**
**(2015). Letters "a" to "r" are used to identify peaks and/or interludes of higher TI/NI.**

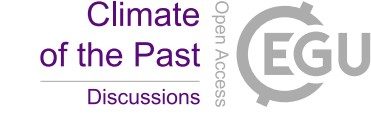



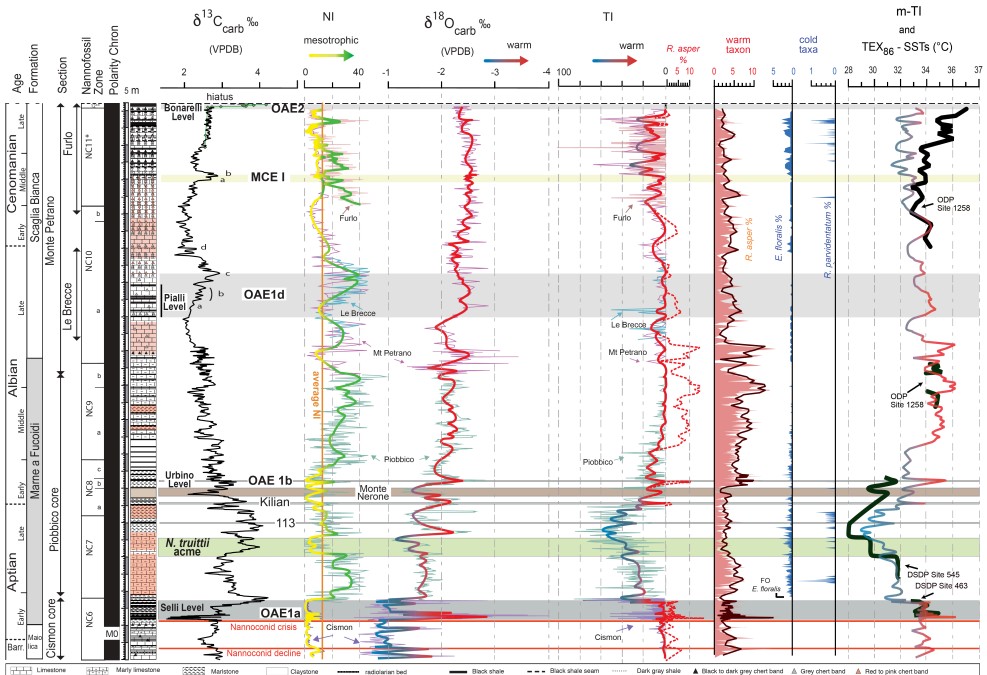


**Figure 6: Composite section through the latest Barremian – Cenomanian which includes from bottom to top: i) Cismon Core, ii) Piobbico Core, iii) Monte Petrano, Le Brecce and Furlo sections. Nannofossil Zones are from Erba (1988), Tiraboschi et al. (2009), Erba et al. (1999), Gambacorta et al., (2015). Smoothed (5pt moving average) δ13C and δ18O curves and point by point δ18O curve include data from the: i) Cismon Core (Bottini et al., 2015), ii) Piobbico Core (Tiraboschi et al., 2009 and Bottini et al., 2015), and**


**iii) Monte Petrano (Giorgioni et al., 2012; Gambacorta et al., 2015). Nannofossil Temperature Index (TI) and Nutrient Index (NI) composite curve are calculated following Bottini et al. (2015) and include nannofossil data from: i) the Cismon Core (Bottini et al., 2015); ii) the Piobbico Core (Tiraboschi et al., 2009; Bottini et al., 2015); iii) the Monte Petrano, Le Brecce and Furlo sections (this work). TI and NI are reported point by point and smoothed (3pt moving average - Monte Petrano, and 5pt moving average - Furlo and Le Brecce). Dotted red line is based on *R. asper* relative abundance. Relative (%) abundance of *R. asper* and cold-water species**


**(*E. floralis* and *R. parvidentatum*) are reported. The NI of Furlo and Monte Petrano are both presented in the composite curve (Fig. 6) since are interpreted to reflect local differences. The modified TI (m-TI) takes into account *R. asper* abundance (%) and it is calibrated with TEX86-SSTs (°C) recalculated following Kim et al. (2010). TEX86-SSTs (black lines) include data from: i) DSDP Site 463 (Schouten et al., 2003); ii) DSDP Site 545 (McAnena et al., 2013); iii) ODP Sites 1258 and 1259 (Forster et al., 2007).**




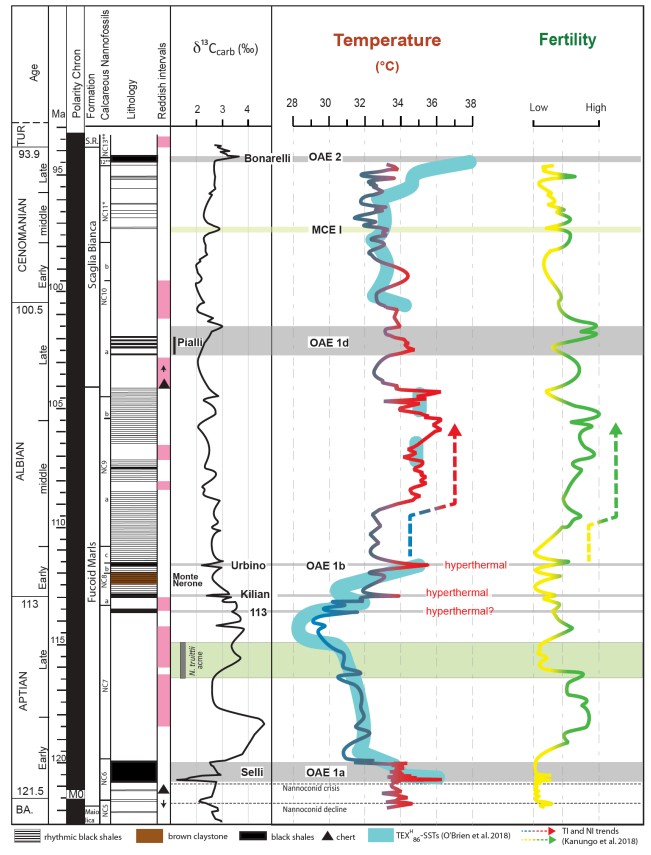

**Figure 7: Synthesis of nannofossil-based nutrient and temperature variations reconstructed in the western Tethys. Time scale is after Malinverno et al. (2010, 2012) for the late Barremian – Aptian and after the Gelogical Time Scale 2012 (Gradstein et al., 2012) for the Albian – Cenomanian. Simplified TEX$^H_{86}$-SSTs for ocean T >15°C (O'Brien et al., 2018) are reported in light blue. Nannofossil TI and NI trends from the Gault Clay Formation (Kanungo et al. 2018) are represented with coloured dotted lines.**
