# Peer review of "Mid-Cretaceous paleoenvironmental changes in the western Tethys"

_Climate of the Past, 2018_

## Referee Comment (RC1) · Anonymous Referee #1 · 2 May 2018

The manuscript "Mid-Cretaceous paleoenvironmental changes in the western Tethys, presented by Bottini and Erba is a very nice example how we can use nannofossil paleoenvironmental indicators to record short- and long-term surface water temperature and fertility changes during the late Barremian to Cenomanian, in particular in sedimentary records where the use of organic and/or inorganic temperature proxies (Mg/Ca, TEX86) is limited. The authors could demonstrate that mid-Cretaceous surface water fertility was mostly fluctuating independently from climatic conditions as well as from black shales intervals with fertility peaks during major Oceanic Anoxic Events. The similarity of western Tethys climatic and fertility fluctuations during OAE 1a, OAE 1b, the middle Albian and OAE 1d from other basins is striking - indicating that the changing conditions are of supraregional significance. I have only minor comments

which might improve the manuscript.

I suggest to use the same scale (percentages, e.g. 0 − 50% or 0 − 25>%) in all figures where you show nannofossil percentages (Fig. 2-4). This gives the reader the possibility to assess which species dominates the signal in which interval.

Line 190: I am a bit unhappy with smoothing the records between Furlo and Le Brecce section differently. Did you use a simple running average or what do you mean with smoothing (e.g., Stineman function, weighted curve fit, . . .)? I suggest treating the records in the same way.

Line 195: "and a few minor cooling intervals (c, d, e?)." The intervals with a "?" indicating cooler intervals. How significant are the presented peaks? I feel that this might be a little bit overinterpreted and I suggest deleting them from the discussion and the figures.

---

## Referee Comment (RC2) · Anonymous Referee #2 · 17 Jun 2018

This is an interesting paper that presents new data as well as incorporating the results of previously published work to construct a continuous section record spanning the mid-Cretaceous (Aptian – Cenomanian). In general, the authors do a fine job in using this record to examine the five goals that they propose in the Introduction, including the overall goal of evaluating the evolution of climate and surface water fertility in the western Tethys region from the mid-Cretaceous. One of the five goals of the paper (presented in the introduction) is to assess the degree of connection of temperature and fertility variations. This is done by the authors in a qualitative manner. This might be expressed better by a simple correlation analysis of all of the data points as well as those from select parts of the sequence (e.g., during hyperthermal intervals). In terms of one of their goals, the correlation of nannofossil proxies with other (geochemical)

paleotemperature proxies, they show a reasonably good correlation (in a qualitative sense). This correlation is best in the Aptian, where the m-TI is directly calibrated with the other paleotemperature proxies, but begins to diverge higher up. Significant divergence of the m-TI and the Tex86 paleotemperature estimates in the upper Cenomanian may reflect the evolutionary changes in nannofossil communities that renders the TI assemblage less useful for paleotemperature estimation. In general, this manuscript needs only minor revision prior to publication, where it will serve as a useful addition to our knowledge.

Some minor specific comments:

Several of the minor points (labeled with letters) in the text and in Figure 5 are apparently defined by only 1-2 data points, strongly suggesting that they may be statistical anomalies, especially given the implied precision of percentages from counts of only 300 specimens. I think that the authors may wish to reconsider these designations and their discussions. The diagrams in Figures 2-4 illustrate the changes in species abundances in the various sections. These diagrams would be easier to interpret if the scales for the species abundances were (mostly) similar. In addition, these diagrams feature a variable called "Total nannofossil abundance (# of Fields of View)". The reader can guess that this represents how many FOV had to be examined to get to a count of "at least 300 specimens", but one is never sure unless the authors specify. Perhaps a sentence in the methods section (2.2) would alleviate this little problem.

Is the raw data going to be supplied with this paper? I saw no mention of how it could be obtained. It would be useful to others in evaluating the paper.

Line 369 & 375 – Blake Nose rather than Black Nose.

―――――――――――――――――

---

## Author Response (AR1)

Dear Luc Beaufort,

We want to thank you for your comments and suggestions that we took under consideration in our review. In the following, we listed the point by point reply to your comments and the replies to Reviewer 1 and 2.
We also added a list of additional minor changes made to the text and the figures.
The revised version of the manuscript was uploaded and includes the changes requested by you and the two Reviewers.

**Reply to the Editor comments**
In general, I found the figures complex, and the captions would increase in readability if the different inserts would be marked with a, b, c, d… or 1, 2, 3….
**REPLY**: we modified figure 5 and 6 accordingly (labelled sections were added to both figures)

In Figure 6: you wrote "The NI of Furlo and Monte Petrano are both presented in the composite curve (Fig. 6) since are interpreted to reflect local differences » Why do you mention (Fig.6) in Fig.6 ? Something is missing between "since" and "are".
**REPLY**: we revised the caption of figure 6 following your suggestions.

Also, in Figure 6: you wrote: "The modified TI (m-TI) takes into account R. asper abundance (%) and it is calibrated with TEX86-SSTs (°C) recalculated following Kim et al. (2010). "Since it is calibrated and since TI is ranging from 0 to 100 and here it goes in temperature, you should change in the figure the Title of the last scale (now m-TI) in C-m-TI or NannoSST or anything you like to mention that it is not in a "TI scale".
**REPLY**: we corrected figure 6 accordingly.

In Figure 7 you should mention in the caption that the SST come from Figure 6 or/and that it is calibrated to TEX… Also the estimates of temperature and fertility are not pointed into the caption when you describe every else. That could confuse a rapid reader who could interpret that your temperature come in fact are from Kanungo et al. This would not be possible if after the first sentence of the caption you write something like "(coloured lines)".
**REPLY**: We changed the representation of Kanungo et al data using a pink line. The figure caption has been updated

**REPLY TO Anonymous Referee #1**

We thank the Reviewer for evaluating our manuscript and providing helpful comments which improved the quality of the revised manuscript. In the following, we list all referee comments and our response.

**RC**: I suggest to use the same scale (percentages, e.g. 0 – 50% or 0 – 25>%) in all figures where you show nannofossil percentages (Fig. 2-4). This gives the reader the possibility to assess which species dominates the signal in which interval.
**REPLY**: We modified figures 2, 3 and 4 accordingly.

**RC:** Line 190: I am a bit unhappy with smoothing the records between Furlo and Le Brecce section differently. Did you use a simple running average or what do you mean with smoothing (e.g., Stineman function, weighted curve fit, …)? I suggest treating the records in the same way.
**REPLY**: In the revised version of the manuscript, the TI and NI of each studied site was smoothed applying a moving average on a different number of data points in order to obtain smoothed TI and NI curves on comparable time-windows. This approach was necessary since: 1) The sedimentation rate changes from the Late Albian to the Cenomanian, 2) The sampling rate differs among the studied sections and, at Furlo, also throughout the section itself.
The new curves take into account the sedimentation rates calculated for the studied sites by Gambacorta et al. (in review). At Le Brecce section, the sampling rate is each 20 cm corresponding to ca. 24 kyrs. The Monte Petrano section was sampled each 50 cm corresponding to ca. 60 kyrs in the Late Albian and to ca. 50 kyrs in the Cenomanian (up to the Bonarelli Level base). At Furlo, the sampling rate is 20 cm corresponding to 20 kyrs in the interval 0-5.8 m and, and ca. 10 cm corresponding to 10 kyrs in the interval 6-30 m.
In order to apply the smoothing to equivalent time-intervals (of ca. 120 kyrs across the Late Albian and of ca. 100 kyrs across the Cenomanian), we calculated a: 3 point moving average at Monte Petrano, a 5 point moving average at Le Brecce, a 6 point moving average in the lower part of the Furlo section (0 to 5.8 m) and a 11 point moving average in the interval 6-30 m of Furlo.
The description of the smoothing method adopted has been revised in chapter 3.3. Figures 5 and 6 were revised and their captions were updated accordingly.
Remarks:
- Line 190 of the Discussion paper: the TI and NI of Monte Petrano were smoothed with a 3 points moving average and not using a 2 point moving average as stated. This part has now been updated in the revised version of the manuscript.
- The new smoothed TI and NI curves show only very minor differences from those presented in the Discussion paper (see revised figures 5 and 6). Consequently, no changes were made to the description of the results and to the Discussion chapter.

**RC**: Line 195: "and a few minor cooling intervals (c, d, e?)." The intervals with a "?" indicating cooler intervals. How significant are the presented peaks? I feel that this might be a little bit overinterpreted and I suggest deleting them from the discussion and the figures.
**REPLY**: A similar comment was made by Reviewer 2. We agree with both Reviewers: most of the intervals were not identifying significant peaks or proper spikes. We have therefore deleted all the letters labelling the TI and NI peaks in figure 5 since some of the labels were not identifying significant peaks (given by one or two single data points) or they were referring to specific temperature/fertility intervals rather than spikes. The text has been modified accordingly (Chapter 3.3.).

**REPLY TO Anonymous Referee #2**

We thank Reviewer 2 for reviewing the manuscript and providing helpful comments that have improved the manuscript. Please see below the point by point response to the specific comments.

**RC**: [omissis…]. One of the five goals of the paper (presented in the introduction) is to assess the degree of connection of temperature and fertility variations. This is done by the authors in a qualitative manner. This might be expressed better by a simple correlation analysis of all of the data points as well as those from select parts of the sequence (e.g., during hyperthermal intervals).
**REPLY**: We calculated the Pearson correlation coefficient between the TI and NI for all studied sites and, additionally, for the onset of OAE 1a, Kilian Level, OAE 1b and OAE 1d. Results show no correlation among the two parameters except during the Kilian Level (r=0.97). The description of the results obtained after the statistical analyses is reported in chapter 4.3.

**RC**: Several of the minor points (labeled with letters) in the text and in Figure 5 are apparently defined by only 1-2 data points, strongly suggesting that they may be statistical anomalies, especially given the implied precision of percentages from counts of only 300 specimens. I think that the authors may wish to reconsider these designations and their discussions.
**REPLY**: We agree with the comment of the Reviewer which finds correspondence with a similar comment by Reviewer 1. As explained in the Reply to Referee 1, we modified Figure 5 by deleting all the letters labelling the TI and NI "peaks" since some of the labels were not identifying significant peaks (given by one or two single data points) or they were referring to specific temperature/ fertility intervals rather than spikes. The text has been modified accordingly (Chapter 3.3.).

**RC:** The diagrams in Figures 2-4 illustrate the changes in species abundances in the various sections. These diagrams would be easier to interpret if the scales for the species abundances were (mostly) similar.
**REPLY**: We modified figures 2, 3 and 4 plotting individual species abundance with the same scale.

**RC**: In addition, these diagrams feature a variable called "Total nannofossil abundance (# of Fields of View)". The reader can guess that this represents how many FOV had to be examined to get to a count of "at least 300 specimens", but one is never sure unless the authors specify. Perhaps a sentence in the methods section (2.2) would alleviate this little problem.
**REPLY**: The total nannofossil abundance represents the average number of nannofossils found in one field of view. We added a brief explanation in chapter 2.2 (line 107-108) as well as in the caption of Figure 2, 3 and 4.

**RC**: Is the raw data going to be supplied with this paper? I saw no mention of how it could be obtained. It would be useful to others in evaluating the paper.
**REPLY**: Calcareous nannofossil data described in this work can be requested contacting the authors. We ask the Editor whether it is, instead, preferred to add the dataset: if so, we can add the tables with the nannofossil data presented in the manuscript as supplementary material.

**RC**: Line 369 & 375 – Blake Nose rather than Black Nose.
**REPLY**: We corrected the text accordingly.

**Further changes made to the text and figures**

- Fig 3 and 5: We added the "fault" symbol which was missing in the Le Brecce section log and added the reference in the text (line 90).
- Figs 2, 3, 4, 5, 6, 7: size font was enlarged where needed.
- Fig 7: we changed the color of the reddish intervals.
- Typing error were corrected throughout the text
- Line 107: we corrected the sample number for Le Brecce and we added the sampling rate for each studied section.

[revised manuscript text omitted]

Furlo and Le Brecce sections

| | | |
|---|---|---|
| **Pagina 6: [2] Eliminato** | **Elisabetta Erba** | **12/07/18 12:30:00** |

of

| | | |
|---|---|---|
| **Pagina 6: [3] Eliminato** | **CB** | **12/07/18 14:14:00** |

.

| | | |
|---|---|---|
| **Pagina 6: [4] Eliminato** | **CB** | **13/07/18 08:54:00** |

apply smoothing

apply smoothing

| Pagina 6: [4] Eliminato | CB | 13/07/18 08:54:00 |
|---|---|---|

apply smoothing

| Pagina 6: [4] Eliminato | CB | 13/07/18 08:54:00 |
|---|---|---|

apply smoothing

| Pagina 6: [4] Eliminato | CB | 13/07/18 08:54:00 |
|---|---|---|

apply smoothing

| Pagina 6: [4] Eliminato | CB | 13/07/18 08:54:00 |
|---|---|---|

apply smoothing

| Pagina 6: [5] Eliminato | Elisabetta Erba | 12/07/18 12:33:00 |
|---|---|---|

| Pagina 6: [5] Eliminato | Elisabetta Erba | 12/07/18 12:33:00 |
|---|---|---|

| Pagina 6: [5] Eliminato | Elisabetta Erba | 12/07/18 12:33:00 |
|---|---|---|

| Pagina 6: [6] Eliminato | CB | 03/07/18 13:31:00 |
|---|---|---|

whilst the TI and NI of Furlo and Le Brecce were smoothed with a 5 point moving average. Relevant peaks (for amplitude and/or stratigraphic correlation) were labelled progressively.

| Pagina 6: [6] Eliminato | CB | 03/07/18 13:31:00 |
|---|---|---|

whilst the TI and NI of Furlo and Le Brecce were smoothed with a 5 point moving average. Relevant peaks (for amplitude and/or stratigraphic correlation) were labelled progressively.

| Pagina 6: [6] Eliminato | CB | 03/07/18 13:31:00 |
|---|---|---|

whilst the TI and NI of Furlo and Le Brecce were smoothed with a 5 point moving average. Relevant peaks (for amplitude and/or stratigraphic correlation) were labelled progressively.

| Pagina 6: [6] Eliminato | CB | 03/07/18 13:31:00 |
|---|---|---|

whilst the TI and NI of Furlo and Le Brecce were smoothed with a 5 point moving average. Relevant peaks (for amplitude and/or stratigraphic correlation) were labelled progressively.

| Pagina 6: [6] Eliminato | CB | 03/07/18 13:31:00 |
|---|---|---|

whilst the TI and NI of Furlo and Le Brecce were smoothed with a 5 point moving average. Relevant peaks (for amplitude and/or stratigraphic correlation) were labelled progressively.

| Pagina 6: [6] Eliminato | CB | 03/07/18 13:31:00 |
|---|---|---|

whilst the TI and NI of Furlo and Le Brecce were smoothed with a 5 point moving average. Relevant peaks (for amplitude and/or stratigraphic correlation) were labelled progressively.

**Pagina 6: [6] Eliminato**          CB          03/07/18 13:31:00

whilst the TI and NI of Furlo and Le Brecce were smoothed with a 5 point moving average. Relevant peaks (for amplitude and/or stratigraphic correlation) were labelled progressively.

**Pagina 6: [7] Eliminato**          CB          03/07/18 13:40:00

(peaks a to d)

**Pagina 6: [8] Eliminato**          CB          03/07/18 13:44:00

(peak b)

**Pagina 6: [8] Eliminato**                    **CB**                              **03/07/18 13:44:00**

(peak b)

**Pagina 6: [9] Eliminato**                    **CB**                              **03/07/18 13:46:00**

(peak d)

**Pagina 6: [9] Eliminato**                    **CB**                              **03/07/18 13:46:00**

(peak d)

**Pagina 6: [9] Eliminato**                    **CB**                              **03/07/18 13:46:00**

(peak d)

**Pagina 6: [10] Eliminato**                   **CB**                              **03/07/18 13:52:00**

the lower interval preceding the MCE I is characterized by

**Pagina 6: [10] Eliminato**                   **CB**                              **03/07/18 13:52:00**

the lower interval preceding the MCE I is characterized by

**Pagina 6: [10] Eliminato**                   **CB**                              **03/07/18 13:52:00**

the lower interval preceding the MCE I is characterized by

**Pagina 6: [10] Eliminato**                   **CB**                              **03/07/18 13:52:00**

the lower interval preceding the MCE I is characterized by

**Pagina 6: [10] Eliminato**                   **CB**                              **03/07/18 13:52:00**

the lower interval preceding the MCE I is characterized by